The emergence of spiraling tracheary element bundles in incompatible grafts

Wu Huiyan
Deng Zhuying
Wang Xue
Liang Dacheng dachengliang@gmail.com
School of Agriculture, Yangtze University, Jingzhou, Hubei, China; Engineering Research Center of Ecology and Agricultural Use of Wetland, Ministry of Education, Yangtze University , Jingzhou , Hubei Province , China
Płachno Bartosz
Electronic publication date: 2022 Sep 14
Publication date: 2022
Volume: 10
Electronic Location ID: e14020
Received 2022 May 16; Accepted 2022 Aug 16
Copyright: ©2022 Wu et al.
Copyright year: 2022
Copyright holder: Wu et al.
License: This is an open access article distributed under the terms of the Creative Commons Attribution License, which permits unrestricted use, distribution, reproduction and adaptation in any medium and for any purpose provided that it is properly attributed. For attribution, the original author(s), title, publication source (PeerJ) and either DOI or URL of the article must be cited.
License URL: https://creativecommons.org/licenses/by/4.0/

Keywords: Interfamilial graft, Xylem, Circular vessels, Spiraling tracheary element, Incompatibility, Portulaca oleracea, Brassica napus, Nicotiana benthamiana, Arabidopsis thaliana

Funding: The authors received no funding for this work.

==============================
In distantly-related plant grafting, incompatibility often occurs between scion and rootstock, resulting in growth stagnation, and eventually graft failure. In this study, we found that an emergent structure, or the spiraling tracheary element (TE) bundles consisting of TE masses occurring at the graft interface, was extensively present in the highly incompatible interfamilial graft of Brassica napus/Portulaca oleracea (Bn/Po) and Nicotiana benthamiana/Portulaca oleracea (Nb/Po). This special structure mostly appeared in the local area near the grafting union, and the frequency and quantity of the spiraling tracheary element bundles were much higher in the scion than in the rootstock. Nevertheless, only a small portion of Arabidopsis thaliana/Portulaca oleracea (At/Po) interfamilial grafts showed a less spiraled TEs at the grafting union (usually a circular TE), which is consistent with its growth performance. This study consolidated that spiraling TE bundles were an important indicator for graft incompatibility. The possible reason for the formation of spiraling TE bundles in interfamilial grafts was discussed.

Introduction

Grafting is one of the artificial propagation methods of plants in which two different plant segments are mechanically joined together to survive as a new individual after vascular connection and wound healing. This technique was practiced in ancient Greece and China as early as the 5th century BC (Melnyk & Meyerowitz, 2015). At present, it is widely used in agricultural production and horticultural research.

The compatibility between scion and rootstock is the key factor affecting the grafting success. A prerequisite for successful grafting and long-term survival is the taxonomic closeness of the scion to the stock. In general, different species of plants in the same genus can form effective union and survive, while grafting of plants in different genera of the same family are rarely compatible (Goldschmidt, 2014). In some special cases of grafting between distant species, such as Arabidopsis and tomato, despite the lack of vascular connection, the Arabidopsis scion grafted onto tomato rootstock can still blossom and produce seeds (Flaishman et al., 2008). However, due to the low efficiency of nutrition and hormone transfer between tissues, it is still doubtful whether these grafts can be maintained for a long time (Melnyk, 2017). So far, the mechanism of the incompatibility of heterograft is still unclear, and practical graft work relies largely on empirical observation for horticultural production.

The connection of vascular bundle after grafting is one of the important signs of grafting success (Melnyk, 2017; Melnyk & Meyerowitz, 2015). The cuts introduced during grafting necessitate the destruction of the vascular system of plants, resulting in the discontinuity of vascular transport of water, nutrients and various organic materials (Asahina & Satoh, 2015). Therefore, the connection of the vascular system between graft partners needs to be quickly re-established for survival. However, various factors including the taxonomic relatedness, anatomical structure (e.g., the necrotic layer at the graft interface), mechanical mismatch during grafting, pathological infections, growth condition and growth activity of either graft partner could cause the graft failure (Hartmann et al., 2011). Mosse (1962) proposed that destruction of vascular continuity due to abnormalities of vascular development at the graft union caused graft incompatibility. In a recent study, formation of a spiraling tracheary element bundle (spiraling TE bundle) in the graft union was strongly associated with graft incompatibility in the Arabidopsis/Nicotiana interfamilial graft (Deng et al., 2021). This further suggests that vascular behavior was critical for graft union formation.

To further test whether this particular vascular structure was occurring in other interfamilial grafts, we focused on interfamilial combination involving purslane as rootstock. The existence of spiraling TEs in the interfamilial grafts, i.e., the Brassica napus/Portulaca oleracea (Bn/Po), Nicotiana benthamiana/Portulaca oleracea (Nb/Po) and Arabidopsis thaliana/Portulaca oleracea (At/Po) combination was described, and the causes of their formation and their influence on the growth of distant grafting were also discussed.

Material and Methods

Experimental materials

The seeds of rape (Brassica napus), Nicotiana benthamiana, Arabidopsis thaliana and purslane (Portulaca oleracea) were surface-sterilized in chlorine gas for 2 h. Then they were plated on the sterile MS medium containing 3% sucrose (W/V). Sterile petri dishes containing seeds were vertically placed in the growth room for constant temperature growth (22∼23 °C), and the growth condition was set at long-day photoperiod cycle (16 h of light and 8 h of darkness).

Micrografting

Seedlings germinated after 7-9 days on MS medium were selected for grafting (purslane as rootstock, rape, Nicotiana benthamiana and Arabidopsis as scion, respectively). The grafting process was described previously (Deng et al., 2021). Grafts were grown on moisturized Whatman paper for 3 days, then gently moved to MS medium containing 3% sucrose (w/v) with forceps and continued to grow in a growth room (16 h light /8 h darkness) at 22−23 °C.

Vasculature isolation, scanning electron microscopy and counting of the spiraling TE bundles

Isolation of vasculature was recently described by Liu et al. (2022) and Deng et al. (2021). The leaves and roots of grafts were removed and soaked in 0.04% saponin solution for 30 min, then washed with PEM solution (50 mM PIPES, 5 mM EGTA, 2.5 mM MgSO4, pH 6.9) three times, and then fixed in 4% paraformaldehyde in PEM buffer for 30 min. The fixed samples were washed with PEM solution and then dissected to remove the tissues adjacent to the vascular bundles under a dissecting microscope. The longitudinally dissected vascular samples were treated with the enzyme cocktail solution consisting of 5 mM 2-(N-morpholino) ethanesulfonic acid (MES), 0.5% (w/v) cellulase (EC 3.2.1.4; Sigma-Aldrich, St. Louis, MO, USA), 0.2% (v/v) pectinase (EC 3.2.1.15; Sigma-Aldrich, St. Louis, MO, USA), 0.12 M sucrose, 1 mM CaCl2(pH 5.5) at 28 °C for 1.5 h. The treated samples were washed in 5% TritonX-100 for 15 min and then in PEM solution for 15 min. Samples were dehydrated step by step (15 min each) with 15%, 30%, 50%, 75% and 100% ethanol. With another two times of absolute ethanol wash, they were dried for 3 h in a −20 °C, using a low-vacuum drier (CHRIST). The dried samples were mounted on the sample holder, and then placed in the ion sputtering apparatus (SC7620 Sputter coater) for conductive gold coating. Examination of the samples was conducted in a VEGA3 TESCAN scanning electron microscope at 20 kv acceleration voltage. In order to count for the number of spiraling bundles, we surveyed the area within a three mm radius centered at the midpoint of the grafting interface. The spiraling bundle was characterized with spiraling tracheary elements plus either a concave hole or convex point, and thus was counted as a spiraling bundle. The Arabidopsis/purslane grafts (n = 67) were surveyed every two days after grafting (DAG), while rape/purslane grafts (n = 35) and Nicotiana/purslane grafts (n = 43) were mainly surveyed after 28 DAG. For each time of collection, graft unions from at least six individual grafts were prepared.

Results

Three interfamilial grafts using purslane as rootstock

In this experiment, we constructed three interfamilial grafts, the Bn/Po heterograft (Figs. 1A and 1B), At/Po heterograft (Figs. 1D and 1E), and Nb/Po heterograft (Figs. 1H and 1I), and four self-grafts, i.e., the Bn (Fig. 1C), At (Fig. 1F), Po (Fig. 1G) and Nb (Fig. 1J) self-graft by micrografting technique. Nearly all the Bn/Po grafts remained quiescent (Fig. 1A), a growth status similar to some grafts of At/Nb combination described previously (Deng et al., 2021), suggesting the two species were highly incompatible. The At/Po grafts, however, grew very quickly at 20 DAG, suggesting that Arabidopsis and purslane was potentially compatible with each other at this stage (Fig. 1D). Usually, a delayed incompatibility was observed around 30 DAG; either the grafts ceased growth, or the grafts developed adventitious roots at the graft union (Table 1). The Nb/Po grafts displayed yellowing of foliage (Fig. 1H, Table 1), suggesting poor development of scion shoot. In addition, the graft unions in both Bn/Po and Nb/Po grafts were enlarged (Figs. 1B and 1I), implying a structural expansion within the unions. In the compatible self-grafts of Bn, At, Po and Nb, the growth of each species was normal (Figs. 1C, 1F, 1G and 1J).

Figure 1 Heterografts of Bn/Po, At/Po and Nb/Po combination and self-grafts of Bn, At, Po and Nb species.

(A) The representative plants of Bn/Po heterografts showing growth stagnation. (B) The graft union of the Bn/Po heterograft. (C) Bn self-graft. (D) A representative plant of At/Po heterograft. (E) The graft union of the At/Po heterograft. (F) At self-graft. (G) Po self-graft. (H) A representative plant of Nb/Po heterograft. (I) The graft union of the Nb/Po heterograft. (J) Nb self-graft. Arrow indicates the graft union.

Table 1 Graft compatibility and occurrence of spiraling bundles.

Grafts	Graft union	Adventitious roots	Retarded growth	Yellowing of foliage	Occurrence of Spiraling bundles	Graft Compatibility	
Bn/Po	Enlarged	17.1%	82.9%	51.4%	80%	Incompatible	
At/Po	Smooth	58.7%	17.4%	54.3%	16.3%	Partially incompatible	
Nb/Po	Enlarged	60.7%	16.6%	100%	64.2%	Highly incompatible	
At/At	Smooth	1%	0	0	0	Fully compatible	
Po/Po	Smooth	0	0	0	0	Fully compatible	
Nb/Nb	Smooth	0	0	0	0	Fully compatible	

Emergence of spiraling TE bundles or circular TE at the graft union

The scion and the rootstock of the Bn/Po heterografts were easily parted during the preparation for SEM, implying fragile connection between scion and rootstock. The SEM examination showed that spiraling TE bundles extensively existed at the graft union of Bn/Po combination (Fig. 2A), and around 80% of grafts produced spiraling TE bundles at the graft union (Fig. 2B). These spiraling bundles were mainly confined to the upper part of grafting interface, and they strongly rejected the connection with TEs from rootstock. A single circular tracheary element could also be formed through self-fusion as shown in Fig. 2A (shown in red circle), and a repeated process of circling led to the formation of spiraling bundles (Fig. 2A). Apparently, these spiraling bundles lost their TE end, and did not provide a tapered or inclined end wall for overlapping or fusion as normal TE did, therefore could not make a connection with the TEs from rootstock.

Figure 2 Occurrence of spiraling TE bundles in interfamilial graft.

(A) A typical Bn/Po graft interface showing the widespread formation of spiraling TE bundles in the scion of Bn/Po graft at 28 DAG. The red circle indicates a single circular TE of rape. (B) Occurrence of various spirals in Bn/Po grafts. Data (mean ± SD) were generated in three repeats (n = 9, 12, and 14 respectively). (C) A small circular TE occurring in the scion of At/Po graft at 15 DAG. (D) A small circular TE (the red circle) occurring in the rootstock of At/Po graft at 27 DAG. (E) Occurrence of various spirals in At/Po grafts. Data (mean ± SD) were generated in three repeats (n = 20, 22, and 25 respectively). (F) The widespread formation of spiraling TE bundles in the scion of Nb/Po graft at 28 DAG. (G) The concave spiral. (H) The convex spiral. (I) The fused spirals were formed into chignon-like sphere structure. (J) Occurrence of various spirals in Nb/Po grafts. Data (mean ± SD) were generated in three repeats (n = 10, 13, and 20 respectively). (K) Occurrence percentage (mean ± SD) of spiraling TEs between At/Po, Nb/Po and Bn/Po grafts. Student’s T-test was used to generate the p-value. An asterisk (*) and two asterisks (**) indicated p < 0.05 and p < 0.01, respectively.

In At/Po grafts, the frequency of spiraling TE bundle formation was relatively low (Figs. 2C and 2D), and about 16.7% of grafts (10 out of 60) harbored circular TEs (Fig. 2E). Moreover, the TEs were loosely spiraled (Fig. 2C), thus they usually existed as a single ring and rarely formed a highly spiraling structure (Figs. 2C and 2D). In fact, the At/Po grafts grew well in the first 20 days, agreeing with the less-forming spiraling bundles.

We further examined the spiraling TEs in the scion of Nicotiana benthamiana belonging to solanaceae family, diverging more than 100 million years ago from brassicaceae family according to Ku et al. (2000). Similarly, the enlarged union was consisted of more spiraling TEs (Fig. 2F) which were either convolute concave (Fig. 2G) or convex (Fig. 2H). In an extreme case, these small spirals were fused to form chignon-like structure (Fig. 2I). More than 65% of these grafts contained various numbers of spirals, of whom the majority were two spirals (Fig. 2J). In all compatible self-grafts, there was no spiraling TE bundle detected (Table 1). Taken together, the more compatible At/Po combination has less spiraling TEs than in Bn/Po and Nb/Po combination (Fig. 2K, Table 1), suggesting a close association of graft incompatibility with emergence of spiraling TEs.

Discussion

In our recent study, we found that the spiraling TE bundles existed in almost all the quiescent grafts of Arabidopsis/Nicotiana combination, but they appeared rarely in the grafts with active growth (Deng et al., 2021). These bundles were most likely built on circular vessels (Deng et al., 2021). The circular vessels were observed in injury-induced wood tumor of Picea excelsa (Lam.) Lk (Włoch, 1976). Similarly, the circular vessels were also formed close to transverse wounds, e.g., the wounds at the inflorescence stems of Arabidopsis (Mazur, Benková & Friml, 2016), at the basal side of the radish root and in basal swellings above the cut surfaces of pea stems about a week after the plant was cut (Sachs, 1981; Sachs & Cohen, 1982). They were also formed at branch junctions of various tree species (Lev-Yadun & Aloni, 1990; Rothwell & Lev-Yadun, 2005), in the suppressed or dormant buds that were oriented across the trunk of Ficus religiosa (Aloni & Wolf, 1984). Usually, these circular structures occurred relatively rare and irregularly in different parts of the plant. The spiraling TE bundles reported here and also by Deng et al. (2021) were widespread at the grafting interface of incompatible grafts. We also observed the occurrence of chignon-like sphere structure consisting of small spirals at the Nb/Po interface (Fig. 2I), which looked like the previously described vascular nodules found in the bud grafting apple tree (Mosse & Labern, 1960). These evidence suggested the formation of the circular vessels might play an important part in grafting process.

Sachs (1984) proposed that the differentiation of circular vessels was related to the circulating flux of auxin. When auxin accumulated near the grafting union, the polarity of signal transmission would be reversed locally with diffusion, resulting in the formation of circular vessels in local areas above the grafting interface. Nevertheless, this explanation might only partially account for the spiraling bundle formation as transverse wounds or injuries usually led to small circular vessels as mentioned above. We did not observe the occurrence of spiraling bundles in the compatible self-grafts of purslane (Table 1), Arabidopsis or Nicotiana (Table 1; Deng et al., 2021), or of quinoa (Liu et al., 2022). The repeated formation of circular vessel at the grafting interface indicated that the spiraling structure could result from the distant scion-rootstock interaction, or alternatively, simply from the emergent property of perturbed auxin flux by heterografting, or both. In addition, it’s easy to comprehend that the spiraling TE bundles could also occur, though in a low frequency, at the rootstock region below the grafting interface where auxin accumulation was much less due to scion-rootstock disconnection.

Conclusions

Spiraling TE bundles existed extensively in incompatible grafts. They restricted the graft growth and potentially blocked the vascular connection, thus could be used an important indicator of incompatibility of distant grafts. In practice, we can use this structure to examine the possible grafting relationship between scion and rootstock. Furthermore, future study could be designed to address the origin and molecular mechanisms of spiraling TE bundles.

Supplemental Information

Supplemental Information 1 Raw data for Fig. 2B, E, J, K

Click here for additional data file.

We would like to thank Dr. Rosemary White for her helpful discussions on this work.

Additional Information and Declarations

Competing Interests

Author Contributions

Data Availability

The authors declare there are no competing interests.

Huiyan Wu performed the experiments, analyzed the data, prepared figures and/or tables, authored or reviewed drafts of the article, and approved the final draft.

Zhuying Deng performed the experiments, authored or reviewed drafts of the article, and approved the final draft.

Xue Wang performed the experiments, prepared figures and/or tables, and approved the final draft.

Dacheng Liang conceived and designed the experiments, analyzed the data, prepared figures and/or tables, authored or reviewed drafts of the article, and approved the final draft.

The following information was supplied regarding data availability:

The raw data is available in the Supplemental File.

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
