# Peer review of "The emergence of spiraling tracheary element bundles in incompatible grafts"

_PeerJ, doi:10.7717/peerj.14020_

## Round 0.1 · original submission · Major Revisions

Dear Authors,

The manuscript is valuable and worth publishing, however, before this, you have to answer all suggestions from Reviewers.

Reviewer 1 ·

Basic reporting

Authors insight in the association between the spiraling tracheary element bundles emergence and graft incompatibility. They studied the highly incompatible interfamilial grafts of Brassica napus/Portulaca oleracea and Nicotiana benthamiana/Portulaca oleracea using micrografting technique. Results were obtained using vasculature isolation protocol and SEM observation. Research topic is very interesting, and the authors approach it with previous expertise. The article is short but concise. The writing is clear, unambiguous, and technically correct. The structure of the article has an acceptable format for PeerJ. However, the format of the references should be revised. For example, scientific names need italics, and some journal names are abbreviated, and others are not.

Introduction has sufficient background. The figures and tables should be improved. Some recommendations to be considered are:

o Images of figure 1 need scale bars. In addition, adding the names of each specie near them in the images could be helpful to the reader.

o o A table with information on the degree of compatibility between each combination based on graft development may be needed to relate it to the other results.

o I recommend using different colors for the bars in the graphs in Figure 2, using the same color for each type of combination. It could be useful for understanding.

Experimental design

Research question is clear in lines 57-61. Methods are well described but are incomplete. The days after grafting when isolation was performed and the number of replicates should appear in the Material and Methods section. The counting method of spiralling tracheary element bundles is also not described in this section.

I think that the experimental design is incomplete. The paper needs to add controls to confirm that the detection of spiraling tracheary element bundles is a consequence of incompatibility in these interfamilial grafts. In other words, the authors need to provide data on the spiraling TEs (%) of at least the purslane autografts.

Validity of the findings

The discussion is of sufficient quality. Conclusions are adequate, are connected to the original research question and are supported by the results if data from at least one control (autograft of purslane) are provided.

Additional comments

A possible typo was detected:
- Line 82 “cellulose (Sigma)” -> “cellulase (Sigma)”

Reviewer 2 ·

Basic reporting

No comment

Experimental design

This MS derives from a previous one from the same group.
In this occasion, authors present a nice MS in which they again identify the formation of spiral tracheal elements at the graft junction of interfamiliar grafts. Moreover they discuss that the spiraling TE bundles in the graft junction is strongly associated with graf incompatibility.
For this purpose they use three experimental systems consisting of interfamilial grafts with a varying degree of incompatibility.
The work is well executed and the results are significant.
However, for the purpose of concluding whether or not spiraling tracheary elements are associated with graft failure, I believe it would be mandatory to show results from a highly compatible graft system such as the Purslane/Purslane autograft.

Validity of the findings

Nothing to add except that the conclusions would be strengthened by showing results from a highly compatible grafting system.

Additional comments

Some minor changes should be also taken into consideration before the MS being accepted for publication:

line 17.- Consider to delete "emergent structure" from abstract.
Line 83.- Replace cellulose by cellulase.
Line 83.- Please, indicate the EC number and source for cellulase and pectinase
Figure 1 caption: I would suggest authors to include a scale bar
Figure 2 legend: Indicate at B, E and J whether data represent the mean and SD.

For the sake of clarity, I would also suggest that the authors modify the figures in Fig. 2B and Fig. 2J by grouping the results of more than 3 spirals into a single category (i.e. "more than 3 spirals").

Reviewer 3 ·

Basic reporting

The manuscript is clearly written. Literature and background are well provided.

In Figure 1 there are no bars in each photo, please add them. I suggest resizing the I photo and equalizing the distances between all photos in both figures. Please title B, E and J in Figure 2 with the names of the heterograft variants.

Please use clearer names of heterografts. For Nicotiana / purslane, Arabidopsis / Nicotiana, Arabidopsis / purslane, N. benthamiana / purslane, replace them with At / Nb and others, respectively.

I suggest using the abbreviation DAG in the text instead of days after the transplant. Please also address them to the photos.

Line 123 checks the names of the families used.

Experimental design

Presented research matches Aims and Scopes of the journal.

Selected methods are described in detail.

Validity of the findings

Results complement the Authors' previous research and bring new interesting data concerning interfamilial grafts.

In conclusions section without any experiments using transport dyes it would be better to state that spiral TE bundles 'potentially' blocked the vascular connection.

Additional comments

no comment

---

## Round 0.2 · accepted · Accept

The authors have satisfactorily addressed all suggestions and questions. I think that now the manuscript is ready for publishing.

Best Wishes,
Bartosz J. Płachno

Reviewer 1 ·

Basic reporting

The authors responded satisfactorily to all my queries. This work could be published as is.

Experimental design

There are no comments.

Validity of the findings

There are no comments.

Additional comments

There are no comments.

Reviewer 2 ·

Basic reporting

No comment

Experimental design

No comment

Validity of the findings

No comment

Additional comments

No comment

Reviewer 3 ·

Basic reporting

I think that the authors have adequately addressed the comments made by all the reviewers in the
revised version of the manuscript. Therefore, I have no further comments.

Experimental design

Scale bars were added to the photos, now it's complete.

Validity of the findings

No comments.

Additional comments

No comments.